# Acyl chain asymmetry and polyunsaturation of brain phospholipids facilitate membrane vesiculation without leakage

**Marco M Manni[1,2], Marion L Tiberti[1], Sophie Pagnotta[3], Hélène Barelli[1], Romain Gautier[1], Bruno Antonny[1]***

[1]Institut de Pharmacologie Moléculaire et Cellulaire, Université Côte d'Azur et CNRS, Valbonne, France; [2]Instituto Biofisika (UPV/EHU, CSIC), Leioa, Spain; [3]Centre Commun de Microscopie Appliquée, Université Côte d'Azur, Nice, France

**Abstract** Phospholipid membranes form cellular barriers but need to be flexible enough to divide by fission. Phospholipids generally contain a saturated fatty acid (FA) at position *sn1* whereas the *sn2*-FA is saturated, monounsaturated or polyunsaturated. Our understanding of the impact of phospholipid unsaturation on membrane flexibility and fission is fragmentary. Here, we provide a comprehensive view of the effects of the FA profile of phospholipids on membrane vesiculation by dynamin and endophilin. Coupled to simulations, this analysis indicates that: (i) phospholipids with two polyunsaturated FAs make membranes prone to vesiculation but highly permeable; (ii) asymmetric *sn1*-saturated-*sn2*-polyunsaturated phospholipids provide a tradeoff between efficient membrane vesiculation and low membrane permeability; (iii) When incorporated into phospholipids, docosahexaenoic acid (DHA; omega-3) makes membranes more deformable than arachidonic acid (omega-6). These results suggest an explanation for the abundance of *sn1*-saturated-*sn2*-DHA phospholipids in synaptic membranes and for the importance of the omega-6/omega-3 ratio on neuronal functions.

DOI: https://doi.org/10.7554/eLife.34394.001

***For correspondence:**
antonny@ipmc.cnrs.fr

**Competing interests:** The authors declare that no competing interests exist.

## Introduction

Although it is common knowledge that polyunsaturated fatty acids (PUFAs) especially omega-3 FAs are important for health, the underlying mechanisms are not fully understood (*Bazinet and Layé, 2014*; *Marszalek and Lodish, 2005*; *Stillwell and Wassall, 2003*). PUFAs act through three different states: as free molecules, as precursors of biological mediators, or as esters in membrane phospholipids. The third form results from the activity of acyl transferases, which selectively incorporate defined fatty acids into phospholipids (*Harayama et al., 2014*; *Shindou et al., 2013*). This allows cells to control the acyl chain profile of their phospholipids, which varies tremendously among organisms, tissues and cells, and even among organelles (*Harayama et al., 2014*; *Hulbert, 2003*; *Shindou et al., 2013*). Interestingly, the FA diversity in phospholipids applies mostly to the *sn2* position of the glycerol backbone, hence resulting in asymmetric phospholipids containing a saturated FA at position *sn1* and an unsaturated FA at position *sn2* (*Hanahan et al., 1960*; *Lands and Merkl, 1963*; *Tattrie, 1959*; *Yabuuchi and O'Brien, 1968*).

For example, the brain is enriched in phospholipids with PUFAs, notably in the case of phosphatidylethanolamine (PE) and phosphatidylserine (PS) (*Tam and Innis, 2006*; *Yabuuchi and O'Brien, 1968*). Moreover, an interesting pattern has been detected in neurons, where the axon tip is enriched in phosphatidylcholine (PC) molecules containing arachidonate (AA or 20:4 omega-6) or

**eLife digest** Surrounding each living cell is a membrane that is mainly made of fat molecules called phospholipids. Similar membranes also surround many of the structures inside cells. It is important for life that these membranes are impermeable to many molecules; for example, they do not allow ions to cross them freely. The membranes also need to be flexible and allow cells to form different shapes. Flexible membranes also allow cells to move molecules around and to divide to produce new cells.

Each phospholipid includes two long chains of atoms called fatty acids. There are many fatty acids but they are typically grouped into saturated and unsaturated based on their chemical structures. The omega-3 and omega-6 fats are both groups of unsaturated fatty acids that are found in brain cells. Many phospholipids in cell membranes contain one saturated and one unsaturated fatty acid but it is not clear why.

By studying fat molecules in the laboratory and combining this with simulations, Manni et al. have now examined the effects of fatty acids on membranes. The investigation showed that phospholipids with both saturated and unsaturated fatty acids strike a balance between impermeable and flexible membranes. More unsaturated fatty acids make more flexible membranes but they are too permeable to be used in cells. The experiments also revealed that omega-3 unsaturated fats aid flexibility more than omega-6. This finding may help to explain why the relative amounts of omega-3 and -6 are so important in the membranes of brain cells.

The connection between the fats we eat and the fatty acids in our cells is complex. Yet, findings like these serve to remind us that we need a balanced diet of different fats to keep all our cells healthy.

DOI: https://doi.org/10.7554/eLife.34394.002

docosahexaenoate (DHA or 22:6 omega-3) at the expense of less unsaturated PC species (*Yang et al., 2012*). Thus, phospholipids with PUFAs are found at very high concentration in synaptic vesicles, where they account for up to 70 mol% of the phospholipid pool (*Takamori et al., 2006*). Retinal discs also show very high concentrations of phospholipids containing PUFAs (*Boesze-Battaglia and Schimmel, 1997*; *Rice et al., 2015*). These striking enrichments suggest that the fatty acyl chain profile of phospholipids could impact on the properties of cellular membranes.

We previously showed that phospholipids with the *sn2* PUFA DHA facilitate the membrane shaping and fission activities of dynamin and endophilin (*Pinot et al., 2014*). These proteins are involved in the formation of endocytic vesicles by assembling into spirals around the neck of membrane buds (*Antonny et al., 2016*; *Boucrot et al., 2015*; *Farsad et al., 2001*; *Slepnev and De Camilli, 2000*; *Sundborger et al., 2011*). Physical manipulations, molecular dynamics simulations and biochemical measurements revealed that DHA-containing phospholipids decrease membrane-bending rigidity by adapting their conformation to membrane curvature, hence providing an advantage for membrane deformation and fission by the dynamin endophilin complex (*Pinot et al., 2014*) in contrast to more rigid membranes, which are less prone to fission (*Morlot et al., 2012*). More generally, the flexibility of polyunsaturated phospholipids along the membrane normal (*z* direction) might soften various mechanical stresses in the membrane (*Barelli and Antonny, 2016*).

The activity of dynamin and endophilin was previously determined on extreme membrane compositions: the phospholipids were either saturated-monounsaturated (16:0-18:1) or saturated-DHA (18:0-22:6ω3) (*Pinot et al., 2014*). However, others PUFAs are found in phospholipids (*Harayama et al., 2014*; *Tam and Innis, 2006*; *Yabuuchi and O'Brien, 1968*). The most common are 18:2 omega-6 (linoleate), 18:3 omega-3 (linolenate), and 20:4 omega-6 (arachidonate), which differ by the number of double bonds and their position along the chain (*Figure 1A*). Here, we present a comprehensive study of the impact of phospholipid unsaturation on the mechanical activities of dynamin and endophilin where we varied both the degree of FA unsaturation and the combination of FAs at position *sn1* and *sn2* of phospholipids considering that most natural phospholipids have an asymmetric FA distribution (*Hanahan et al., 1960*; *Lands and Merkl, 1963*; *Tattrie, 1959*; *Yabuuchi and O'Brien, 1968*). The analysis reveals that the combination of an *sn1* saturated acyl

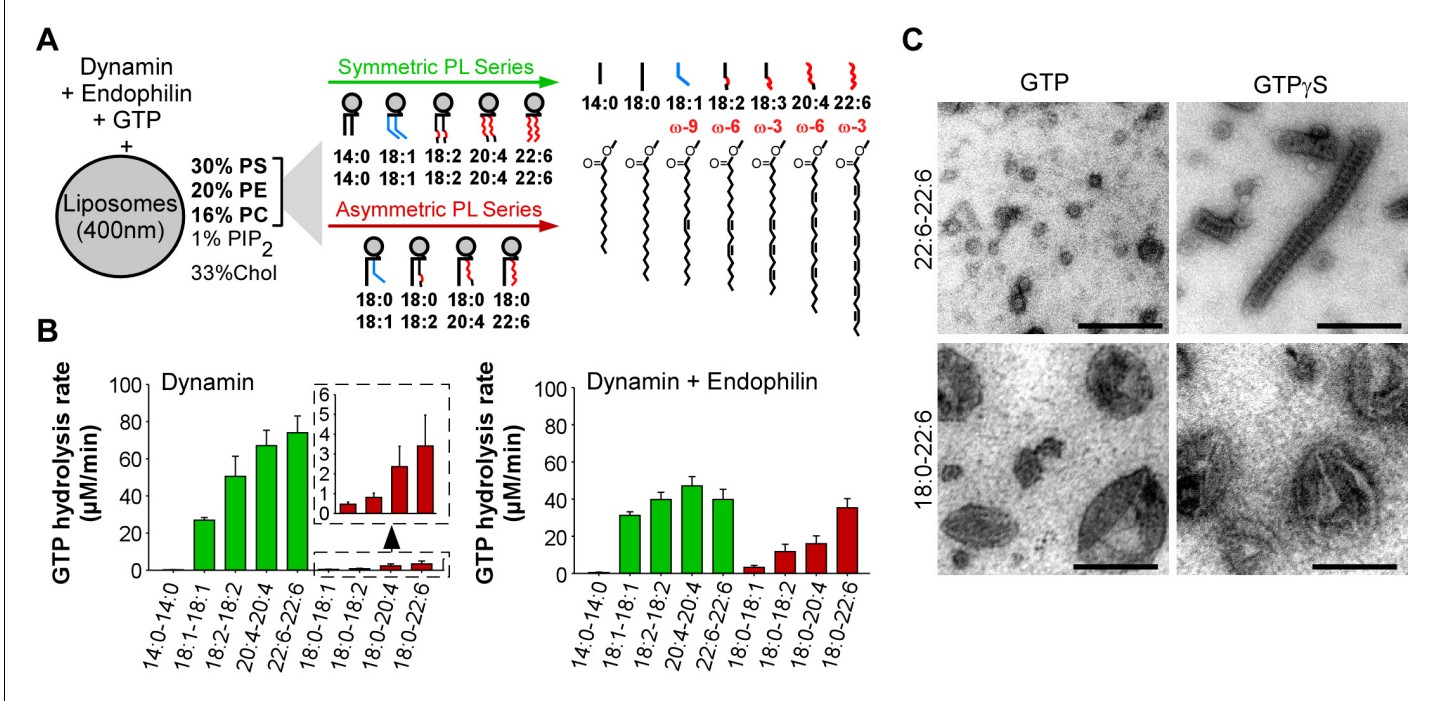

**Figure 1.** GTPase activity of dynamin on liposomes made of phospholipids with different combinations of acyl chains. (A) Principle of the experiments and chemical structure of the various phospholipid species used in this study. (B) Rate of GTP hydrolysis by dynamin (0.3 µM) ±endophilin (0.6 µM) with large liposomes (400 nm extrusion) containing phospholipids with symmetric (green) or asymmetric (red) acyl chains at positions *sn1* and *sn2* as shown in A. Data are mean ± SD from three independent experiments. All lipid compositions are detailed in **Supplementary file 1**. (C) Electron microscopy analysis of 22:6-22:6 or 18:0-22:6 liposomes after incubation with dynamin (0.5 µM) and with GTP or GTPγS. With GTPγS, numerous dynamin spirals formed on 22:6-22:6 liposomes but not on 18:0-22:6 liposomes. With GTP, almost all 22:6-22:6 liposomes were transformed into very small (radius <20 nm) structures, whereas large parental profiles were still abundant in the case of 18:0-22:6 liposomes. Scale bar, 200 nm.

DOI: https://doi.org/10.7554/eLife.34394.003

The following figure supplement is available for figure 1:

**Figure supplement 1.** Time course of GTP hydrolysis by dynamin in the presence of phospholipids with different acyl chains and EM analysis.

DOI: https://doi.org/10.7554/eLife.34394.004

chain and an *sn2* polyunsaturated acyl chain solves the conundrum between making a membrane very permissive to vesiculation while maintaining a proper control of membrane permeability.

## Results

### Comprehensive analysis of the effect of phospholipid unsaturation on dynamin GTPase activity

GTP hydrolysis in the dynamin spiral occurs by a mutual nucleophilic attack between dynamin molecules from adjacent rungs (*Chappie et al., 2011*). Consequently, the rate of GTP hydrolysis depends on dynamin self-assembly and, in effect, increases from negligible values in solution where dynamin is not polymerized, to rates in the range of 2 to 5 s$^{-1}$ on optimal membrane templates where dynamin forms spirals (*Stowell et al., 1999*). We reasoned that GTPase measurements should provide a robust, although indirect, assay to survey a comprehensive library of liposomes made of phospholipids of defined acyl chains for their permissibility to the mechanical activity of dynamin. Dynamin was purified from rat brain, which contains mostly the neuron-specific dynamin-1 isoform, which has a higher membrane curvature generating activity than dynamin-2, the ubiquitous isoform (*Liu et al., 2011*). This screen could also be performed in the presence of proteins that cooperate with dynamin (e.g. BAR domains). Thereafter, the most interesting membrane parameters could be further analyzed by more direct assays of dynamin mechanical activity (e.g. EM observations or assays with Giant Unilamellar Vesicles (GUVs)), which are difficult to standardize for large screens. This second

round of analysis is important because conditions exist where dynamin readily self-assembles and undergoes fast GTP hydrolysis and yet does not efficiently promote membrane vesiculation (*Neumann and Schmid, 2013*; *Stowell et al., 1999*).

We prepared large unilamellar vesicles (extrusion 400 nm) made of five lipids: PC, PE, PS, phosphatidylinositol(4,5)bisphosphate (PI(4,5)P$_2$) and cholesterol (*Figure 1A*). The relative amount of these lipids was kept constant and was chosen to be compatible with the recruitment of both dynamin, which interacts with PI(4,5)P$_2$, and of BAR domain proteins, which interacts with negatively charged lipids (e.g. PS and PI(4,5)P$_2$). However, PI(4,5)P$_2$ was present at low density (1 mol %; close to physiological values) to amplify the need for other facilitating factors such as cooperation with endophilin and membrane flexibility. The only variable in the liposome formation was the acyl chain profile of PC, PE and PS, which accounted for 99% of total phospholipids. Using commercially available or custom lipids, we systematically changed the acyl chain profile in two ways (*Supplementary file 1* and *Figure 1A*). First, we gradually increased the length and unsaturation level of both the *sn1* and *sn2* acyl chains of PC, PE and PS according to the series 14:0-14:0, 18:1-18:1 (omega-9), 18:2-18:2 (omega-6), 20:4-20:4 (omega-6), and 22:6-22:6 (omega-3). Considering that most physiological phospholipids have different acyl chains at positions *sn1* and *sn2* (*Hanahan et al., 1960*; *Lands and Merkl, 1963*; *Tattrie, 1959*; *Yabuuchi and O'Brien, 1968*), we performed a second series in which we maintained a saturated (18:0) acyl chain at position *sn1* and solely changed the *sn2* chain (18:0-18:1, 18:0-18:2, 18:0-20:4, 18:0-22:6). These asymmetric combinations are most frequent in mammalian lipids. The parallel increase in acyl chain length and unsaturation enabled all lipid mixtures to be fluid above 20°C (*Huang, 2001*).

The result of this comprehensive analysis is shown in *Figure 1B* (typical GTPase experiments are provided in *Figure 1—figure supplement 1A*). Despite the identical composition of all liposomes in term of lipid polar head groups, the rate of GTP hydrolysis by dynamin varied up to 300-fold indicating that dynamin is very sensitive to the acyl chain content of the lipid membrane on which it acts. Two parameters emerged: acyl chain asymmetry and acyl chain unsaturation. First, the GTPase activity of dynamin increased dramatically (x 20) on membranes made of symmetric diunsaturated phospholipids compared to asymmetric saturated-unsaturated phospholipids. Second, the GTPase activity of dynamin increased with the unsaturated level of phospholipids (18:1 < 18:2 < 20:4 < 22:6), a trend that was observed in both symmetric and asymmetric phospholipid series.

## Membranes with symmetric polyunsaturated phospholipids are highly permeable to solutes

How to explain the spectacular effect of symmetric diunsaturated phospholipids on the GTPase activity of dynamin? By negative staining electron microscopy, we observed that dynamin alone extensively deformed 22:6-22:6 liposomes, whereas 18:0-22:6 liposomes were largely unaffected (*Figure 1C* and *Figure 1—figure supplement 1B*). The analysis was performed either in the presence of GTPγS, where dynamin self-assembles into stable spirals on membranes, or in the presence of GTP, where dynamin spirals further constrict to promote liposome vesiculation into round profiles of ≈ 20 nm in diameter. Liposomes containing phospholipids with two polyunsaturated acyl chains appeared exceptionally malleable as compared to liposomes made of phospholipids with one saturated and one polyunsaturated acyl chain.

Deformation of spherical liposomes is necessarily accompanied by a diminution of volume of the encapsulated solution. For example, the COPI coat is more efficient at making vesicles from liposomes that have been permeabilized with a pore-forming toxin (*Manneville et al., 2008*). Because a previous study reported that membranes made of 18:2-18:2-PC or 18:3-18:3 PC showed a two to three fold higher water permeability than membranes made of 18:0-18:1-PC or 18:0-18:2-PC (*Olbrich et al., 2000*), we suspected that membranes with dipolyunsaturated phospholipids might be very permissive to deformation by dynamin due to higher permeability.

To assess the permeability of our artificial membranes, we combined molecular dynamics simulations with various measurements. The simulations were performed at the all-atom scale on bilayers containing 2 × 144 phospholipids with the same composition as that used in the experiments. To evaluate water permeability, we determined the number of water molecules that visit the hydrophobic part of the membrane during a period of 100 ns (*Figure 2A*). These movements were subdivided into two classes: events in which the water molecule fully crosses the lipid bilayer; events in which the water molecule enters into the hydrophobic region of the bilayer and then exists on the same

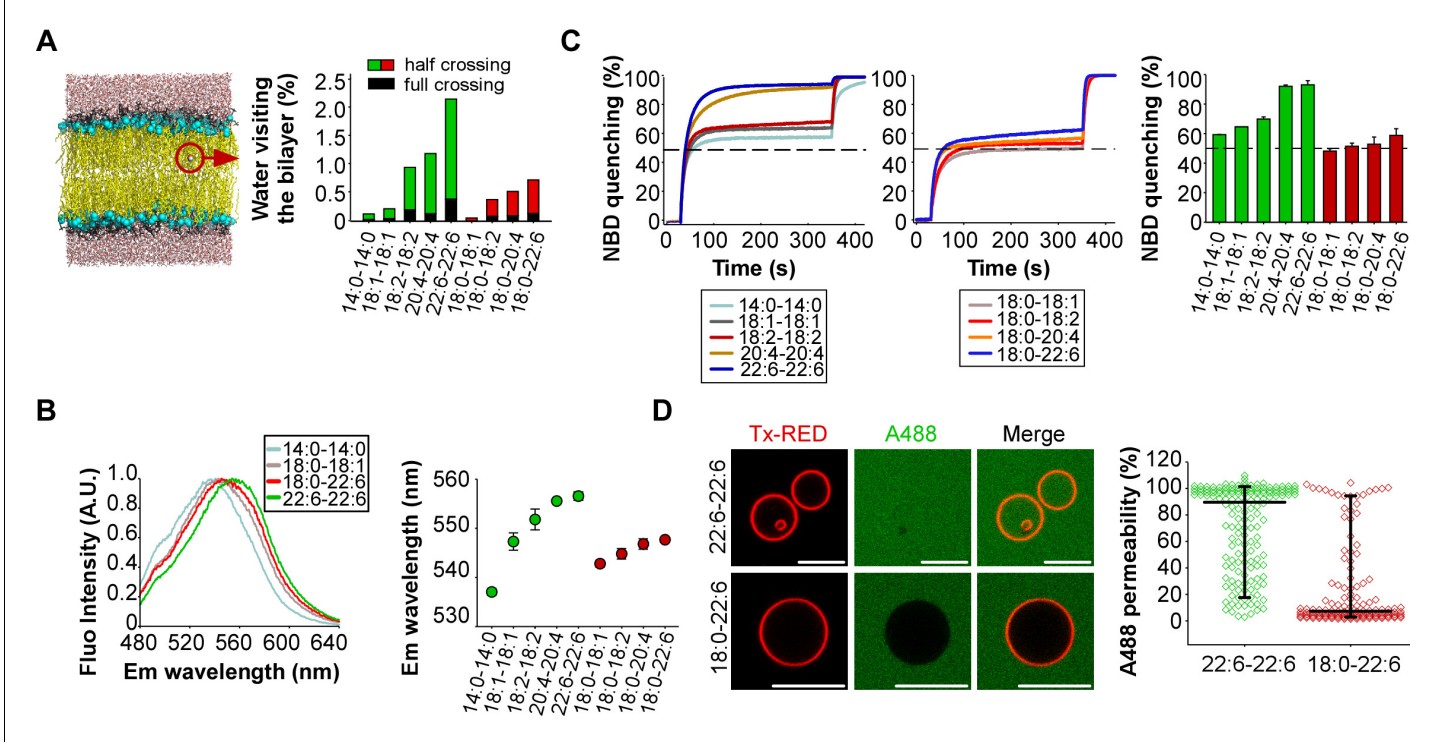

**Figure 2.** Membranes with symmetrical but not asymmetrical polyunsaturated phospholipids are highly permeable. (**A**) Left: snapshot of a lipid bilayer with the same composition as that used in the dynamin GTPase assay. The red circle highlights a water molecule in the membrane hydrophobic region. Right: % of water molecules visiting the membrane hydrophobic region during a 100 ns simulation as a function of phospholipid acyl chain composition. (**B**) Hydratation of the interfacial region of the same liposomes as that used in *Figure 1* as measured by the fluorescence of the push-pull pyrene dye PA. Left: typical emission fluorescence spectra. Right: emission wavelength as a function of phospholipid acyl chain composition. (**C**) Dithionite-mediated NBD quenching assay. At t = 30 s, dithionite was added to the same liposomes as that used in the GTPase assay but also containing PE-NBD. The dotted line indicates 50% of quenching, which is the expected value if only externally orientated PE-NBD molecules are quenched by dithionite. At t = 350 s, 0.1% Triton X-100 was added to allow the quenching of all PE-NBD molecules. Data on the right bar plot are mean ±variation from two independent experiments. (**D**) GUVs permeability measurements. GUVs containing the 55 mol % of the indicated polyunsaturated phospholipids were incubated with soluble Alexa488 and imaged by confocal microscopy. Data show all GUVs measured from six independent experiments and from two independent GUV preparations (large horizontal bar: mean; vertical bar SD). All lipid compositions are detailed in *Supplementary file 1*.

DOI: https://doi.org/10.7554/eLife.34394.005

The following figure supplement is available for figure 2:

**Figure supplement 1.** Emission fluorescence spectrum of the push-pull pyrene dye PA.

DOI: https://doi.org/10.7554/eLife.34394.006

side. In saturated (14:0-14:0) membranes, only few water movements were detected. In membranes made of phospholipids with one unsaturated FA, the number of moving water molecules increased up to 10-fold with the level of acyl chain unsaturation (18:1 < 18:2 < 20:4 < 22:6) (*Figure 2A*). Importantly, membranes with phospholipids containing two unsaturated FA showed a further 2 to 3-fold increase in the number of moving water molecules as compared to membranes with asymmetric saturated-unsaturated phospholipids. This increase occurred whatever the acyl chain considered (e.g. 20:4-20:4 *vs* 18:0-20:4 or 22:6-22:6 *vs* 18:0-22:6).

If membranes with symmetric polyunsaturated phospholipids are more hydrated than membranes with asymmetric saturated-unsaturated phospholipids, this should influence the fluorescence of polarity-sensitive dyes at the membrane interface. To investigate this possibility, we used a recently synthesized push-pull pyrene (PA). This probe, similarly to the popular probe Laurdan, changes its emission maximum as a function of membrane hydration and solvent relaxation, which are parameters linked to lipid order (*Niko et al., 2016*). PA showed a gradual red shift in emission when the number of double bonds in the *sn2* acyl chain increased (18:0-18:1 < 18:0-18:2 < 18:0-20:4 < 18:0-

22:6) (*Figure 2B* and *Figure 2—figure supplement 1*). However, replacing asymmetric saturated-polyunsaturated phospholipids with dipolyunsaturated phospholipids caused a much larger red shift (*Figure 2B* and *Figure 2—figure supplement 1*; e.g. 18:0-22:6 << 22:6-22:6), suggesting that duplication of the PUFA in phospholipids dramatically increased membrane hydration.

Considering the importance of maintaining ion gradients across biological membranes, we next assessed the permeability of our liposomes to the oxoanion dithionite ($[S_2O_4]^{2-}$) by an NBD quenching assay. When added to liposomes, dithionite (MW = 128 Da) immediately quenches the fraction of (C16:0-C16:0) PE-NBD that is present in the outer leaflet. This fraction is about 50% but varies depending on factors like membrane curvature or the presence of multi-lamellar liposomes (*Kamal et al., 2009*). Thereafter, dithionite slowly quenches the remaining PE-NBD molecules. This process occurs either by penetration of dithionite into the liposomes or because of PE-NBD flip-flop. Previous work established that dithionite entry is about 1000 times faster than lipid flip-flop (*Armstrong et al., 2003*). Therefore, the slow phase in PE-NBD quenching experiments should reflect dithionite permeability. *Figure 2C* shows that membrane permeability to dithionite modestly increased with the level of polyunsaturation in asymmetric phospholipids (18:0-18:1 < 18:0-18:2 < 18:0-20:4 < 18:0-22:6) and that liposomes with symmetric polyunsaturated phospholipids showed a > 10 fold higher permeability. This effect was particularly evident for 20:4-20:4 and 22:6-22:6 membranes: 95% of PE-NBD was quenched after 300 s incubation with dithionite as compared to 55–60% in the case of 20:4 and 18:0-22:6 membranes. Note that the liposomes used in the dithionite experiments were the same as that used in the dynamin experiments (see *Figure 1B*) to allow a direct comparison between the two assays. However, a drawback of liposomes obtained by extrusion through large pore size filters (here 400 nm), is the presence of multi-lamellar species (*Kamal et al., 2009*). For such species, dithionite has to cross several bilayers to fully quench all PE-NBD molecules. This effect probably explains why the second phase of PE-NBD quenching, although quite fast in the case of 20:4-20:4 and 22:6-22:6 liposomes, was not complete; a small percentage ($\approx$ 5%) of NBD signal remained unquenched after 300 s incubation (*Figure 2C*). Despite these limitations, these experiments suggest that the presence of two polyunsaturated acyl chains in phospholipids strongly compromise membrane impermeability to ions.

Last, we visualized the permeability of GUVs to the large fluorescent solute Alexa A488 maleimide (MW = 720 Da). This compound was added externally to the GUVs, which were imaged by fluorescence microscopy (*Figure 2D*). Again, the difference between symmetric and asymmetric polyunsaturated phospholipids was clear-cut. GUVs containing 55 mol% 22:6-22:6 phospholipids were about 10 times more permeable to Alexa A488 maleimide than GUVs containing 55 mol% 18:0-22:6 phospholipids.

Altogether, these experiments revealed a remarkable correlation between the ability of dynamin alone to readily vesiculate membranes and the permeability of these membranes to water and even to large or charged solutes.

## Dynamin GTPase activity on asymmetric saturated-polyunsaturated phospholipids

If membranes with symmetric diunsaturated phospholipids appear exceptionally prone to vesiculation by dynamin, their high permeability to ions and large solutes disqualify them for the formation of selective membrane barriers. The GTPase assay of *Figure 1B* and the membrane permeability experiments of *Figure 2* suggest that asymmetric saturated-polyunsaturated phospholipids offer a compromise between low permeability and dynamin activity. Even though the intrinsic GTPase activity of dynamin on such liposomes was low, it increased about 10 times in the presence endophilin-A1 (*Figure 1B*).

In cells, endophilin works in close partnership with dynamin both through protein/protein interactions and through the ability of the two proteins to form membrane-deforming spirals (*Boucrot et al., 2015*; *Farsad et al., 2001*; *Sundborger et al., 2011*). The BAR domain of endophilin is followed by an SH3 domain, which interacts with the proline-rich region of dynamin leading to cooperative BAR/dynamin membrane recruitment (*Farsad et al., 2001*; *Meinecke et al., 2013*; *Sundborger et al., 2011*). In addition, membrane deformation by BAR domains facilitates dynamin self-assembly, which by itself is a relatively weak membrane deforming protein and which preferentially self-assembles on pre-curved membranes (*Neumann and Schmid, 2013*; *Roux et al., 2010*),

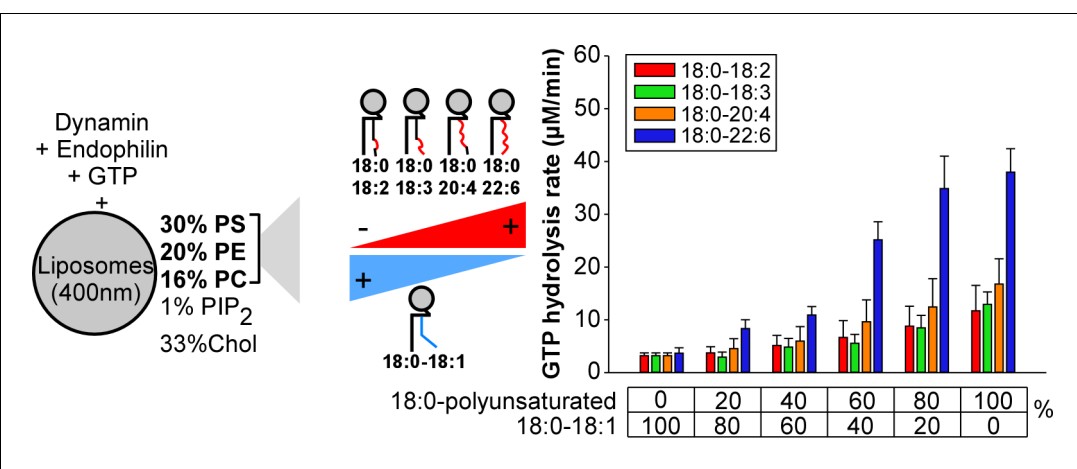

**Figure 3.** Higher activity of dynamin on liposomes containing 18:0-22:6 phospholipids as compared to other polyunsaturated phospholipids. GTPase activity of dynamin (0.3 μM) in the presence of endophilin (0.6 μM) and with large liposomes (400 nm extrusion) containing increasing amounts of the indicated asymmetric polyunsaturated phospholipids at the expense of 18:0-18:1 phospholipids. Data are mean ±SD from three independent experiments. All lipid compositions are shown in *Supplementary file 1*.

DOI: https://doi.org/10.7554/eLife.34394.007

The following figure supplement is available for figure 3:

**Figure supplement 1.** Dynamin activity at different $PIP_2$ levels or with different BAR domains on polyunsaturated membranes.

DOI: https://doi.org/10.7554/eLife.34394.008

unless the membranes are particularly deformable as observed with symmetric unsaturated phospholipids (*Figure 1C*).

These considerations and the fact that asymmetric polyunsaturated phospholipids are much more frequent than symmetric ones in biological membranes prompted us to focus on reconstitutions in which both dynamin and endophilin were present and acted on membranes with asymmetric phospholipids. Under such conditions, the exact nature of the *sn2* acyl chain appeared very important: significant differences in the dynamin GTPase activity were observed between acyl chain combinations that are chemically very close (e.g. 18:0-20:4 *vs* 18:0-22:6) (*Figure 1A*).

To better analyze these differences, we repeated the GTPase assay under conditions where we gradually increased the amount of asymmetric polyunsaturated phospholipids at the expense of 18:0-18:1 phospholipids (*Figure 3*). Two omega-3 combinations (18:0-18:3 and 18:0-22:6) and two omega-6 combinations (18:0-18:2 and 18:0-20:4) were included in the analysis to evaluate the importance of the double bond position. All polyunsaturated phospholipids facilitate dynamin GTPase activity. However, the effect of 18:0-22:6 phospholipids surpassed by 2 to 4-fold that observed with the less complex asymmetric polyunsaturated phospholipids (18:0-18:2, 18:0-18:3 and 18:0-20:4) (*Figure 3*). Dynamin GTPase activity plateau at about 60 mol% of 18:0-22:6 phospholipids, close to the amount of polyunsaturated lipids in synaptic vesicles (*Takamori et al., 2006*).

To check that the effect of asymmetric polyunsaturated phospholipids was not restricted to our particular conditions, we modified several parameters. First, we varied the % of PI(4,5)P$_2$. In a background of 18:0-18:1 phospholipids, the GTPase activity of dynamin in the presence of endophilin was low and increased with the % of PI(4,5)P$_2$ (from 0% to 5%; *Figure 3—figure supplement 1A*). In a background of 18:0-18:2, 18:0-18:3 or 18:0-20:4 phospholipids, the GTPase activity was much higher and required not more than 1 mol% PI(4,5)P$_2$. Strikingly, the activity of dynamin was almost maximal with 18:0-22:6 phospholipids even in the absence of PI(4,5)P$_2$ (*Figure 3—figure supplement 1A*). Next, we replaced endophilin by SNX9, another BAR-domain containing protein that interacts with dynamin. Both endophilin and SNX9 increased dynamin activity much more efficiently on membranes containing 18:0-22:6 phospholipids or 18:0-20:4 than 18:0-18:1 phospholipids (*Figure 3—figure supplement 1B*). In addition, SNX9 was more efficient than endophilin for assisting dynamin activity in agreement with a previous study (*Neumann and Schmid, 2013*). All these

experiments converge towards the same conclusions. First, all asymmetric saturated-polyunsaturated phospholipids favor dynamin GTPase activity, notably under conditions close to physiological conditions (low concentration of PI(4,5)P$_2$, low protein concentration, cooperation with BAR-domain proteins). Second, docosahexaenoic acid (22:6), which is the most polyunsaturated species of the omega-3 family, surpasses all other tested species including arachidonate (20:4), the most polyunsaturated species of the omega-6 family.

## Membrane fission by dynamin and endophilin is sensitive to the omega-6/omega-3 ratio

18:0-20:4 and 18:0-22:6 phospholipids are abundant in specialized membranes (e.g. synaptic vesicles; *Takamori et al., 2006*). Considering the importance of the omega-6/omega-3 ratio for health, we next focused on these acyl chain combinations and used 18:0-18:1 membranes as negative control.

By transmission electron microscopy (TEM) we observed that both 18:0-20:4 and 18:0-22:6 liposomes but not 18:0-18:1 liposomes, became extensively deformed after incubation with dynamin, endophilin and GTPγS or GTP. With GTPγS, membrane tubulation dominated (*Figure 4A*). The tubes were surrounded by a protein spiral with a pitch of ~20 nm characteristic of the endophilin-dynamin complex (*Farsad et al., 2001*; *Pinot et al., 2014*; *Sundborger et al., 2011*) (*Figure 4B* and *Figure 4—figure supplement 1A*). However, the tubes formed from 18:0-22:6 membranes were significantly thinner than the tubes formed from 18:0-20:4 membranes (*Figure 4B*). With GTP present, liposome vesiculation dominated (*Figure 4A*). The size distribution of the membrane profiles was different between 18:0-20:4 and 18:0-22:6 phospholipids (*Figure 4C*). With 18:0-20:4 phospholipids, there was a remaining peak of large membrane profiles (R > 50 nm), which coexisted with a peak of small vesicles (R < 50 nm). With 18:0-22:6 phospholipids, the liposomes were almost fully transformed into small vesicles. In addition, the vesicles formed from 18:0-22:6 membranes were slightly smaller than that formed from 18:0-20:4 membranes (*Figure 4C*). Note that after short incubation with GTP, some tubes were observed both with 18:0-20:4 and 18:0-22:6 liposomes (*Figure 4—figure supplement 1B*). These tubes were not straight as with GTPγS but showed constrictions, suggesting snapshots in the process of membrane fission (black arrows in *Figure 4—figure supplement 1B*)

Considering the technical limitations caused by spontaneous membrane fission on TEM grids (*Danino et al., 2004*), we next performed a GUV shrinking assay (*Meinecke et al., 2013*). In these experiments, dynamin, endophilin and GTP were added to GUVs containing 55 mol% of polyunsaturated phospholipids, which were pre-stabilized in buffer at the reaction temperature (37°C). Dynamin, endophilin and GTP caused GUV consumption over time for both 18:0-20:4 and 18:0-22:6 membranes (*Figure 5A*) but not for the control GUVs that contained only 18:0-18:1 phospholipids (*Figure 5—figure supplement 1A and B*). After 1 hr of incubation, the difference between 18:0-20:4 and 18:0 22:6 phospholipids was significant, as we detected a larger population of shrunk 18:0-22:6 GUVs and a higher amount of intact 18:0-20:4 GUVs (*Figure 5B*). The difference in GUV shrinking between 18:0-20:4 and 18:0-22:6 phospholipids was already evident after 15 min and increased over time (*Figure 5C*).

## Differences between asymmetric polyunsaturated phospholipids as captured by MD simulations

To better understand the advantage provided by asymmetric polyunsaturated phospholipids on membrane deformation and fission, we conducted MD simulations on lipid bilayers. In the coarse-grained mode, we considered large membrane patches and imposed a pulling force to deform them into a tube, which might undergo fission (*Baoukina et al., 2012*; *Pinot et al., 2014*) (*Figure 6A and B*). This approach is informative in terms of membrane mechanics, but the 1:4 scale of the MARTINI force field (one elementary bead for 3 to 4 bonded atoms) makes the depiction of the chemistry of polyunsaturated phospholipids quite imprecise. Nevertheless, we could construct PC bilayers in which acyl chains made of 4 or five beads approximate the series 18:0-18:1, 18:0-18:2, 18:0-20:4 and 18:0-22:6 (*Figure 6A and B*). In the all-atom mode, we considered membrane patches of 2 × 144 phospholipids with the same composition as that used in the experiments. This approach is limited

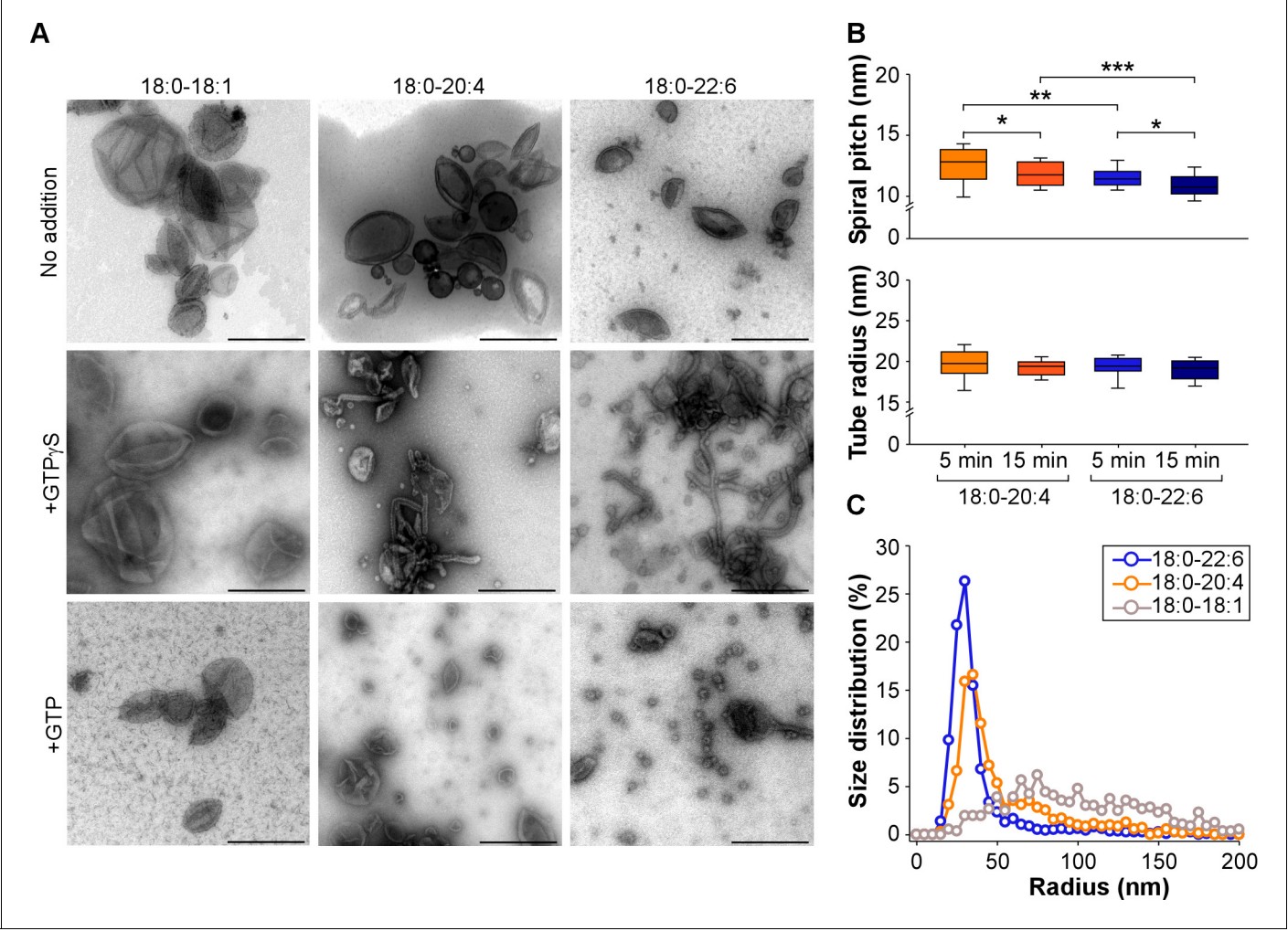

**Figure 4.** Higher vesiculation activity of dynamin and endophilin on liposomes containing 18:0-22:6 phospholipids compared to 18:0-20:4 phospholipids. (A) Electron micrographs of 18:0-18:1, 18:0-20:4 and 18:0-22:6 liposomes (400 nm extrusion) before or after incubation with dynamin (0.5 µM), endophilin (1 µM), and GTP or GTPγS (500 µM). Scale bar, 500 nm. (B–C) Quantification of membrane tubulation (B) and vesiculation (C) from three independent experiments similar to that shown in A. In B, the tube radius and the protein spiral pitch was determined after 5 or 15 min incubations of the liposomes with dynamin, endophilin and GTPγ (see also *Figure 4—figure supplement 1*). In C, the size distribution of the membrane profiles was determined after 30 min incubation of the liposomes with dynamin, endophilin and GTP. All lipid compositions are showed in *Supplementary file 1*.

DOI: https://doi.org/10.7554/eLife.34394.009

The following figure supplement is available for figure 4:

**Figure supplement 1.** EM analysis of the dynamin, endophilin and liposome mixtures.

DOI: https://doi.org/10.7554/eLife.34394.010

to flat membranes but enables an accurate description of the slight chemical differences between polyunsaturated acyl chains (*Figure 6C–E*).

For all coarse-grained membranes tested, applying a constant force above a threshold of 175 kJ $mol^{-1}$ $nm^{-1}$ induced the formation of a tube, which grew by a fast protrusion phase followed by a linear phase as previously observed (*Baoukina et al., 2012*). Increasing the degree of phospholipid polyunsaturation (18:0-18:1 < 18:0-18:2 < 18:0-20:4 < 18:0-22:6) accelerated the linear phase and resulted in the formation of longer and thinner tubes (*Figure 6A* and *Figure 6—figure supplement 1A and B*). Because the bending energy of a membrane tube is proportional to the ratio between tube length and radius ($E_b = \pi K_b L/R$), we plotted $L/R$ as a function of the applied force (*Figure 6A*). At t = 200 ns and for $F = 200$ KJ $mol^{-1}$ $nm^{-1}$, $L/R$ = 6, 12, 16 and 20 nm/nm for 18:0-18:1, 18:0-18:2, 18:0-20:4 and 18:0-22:6 tubes, respectively. Considering that these tubes should have stored

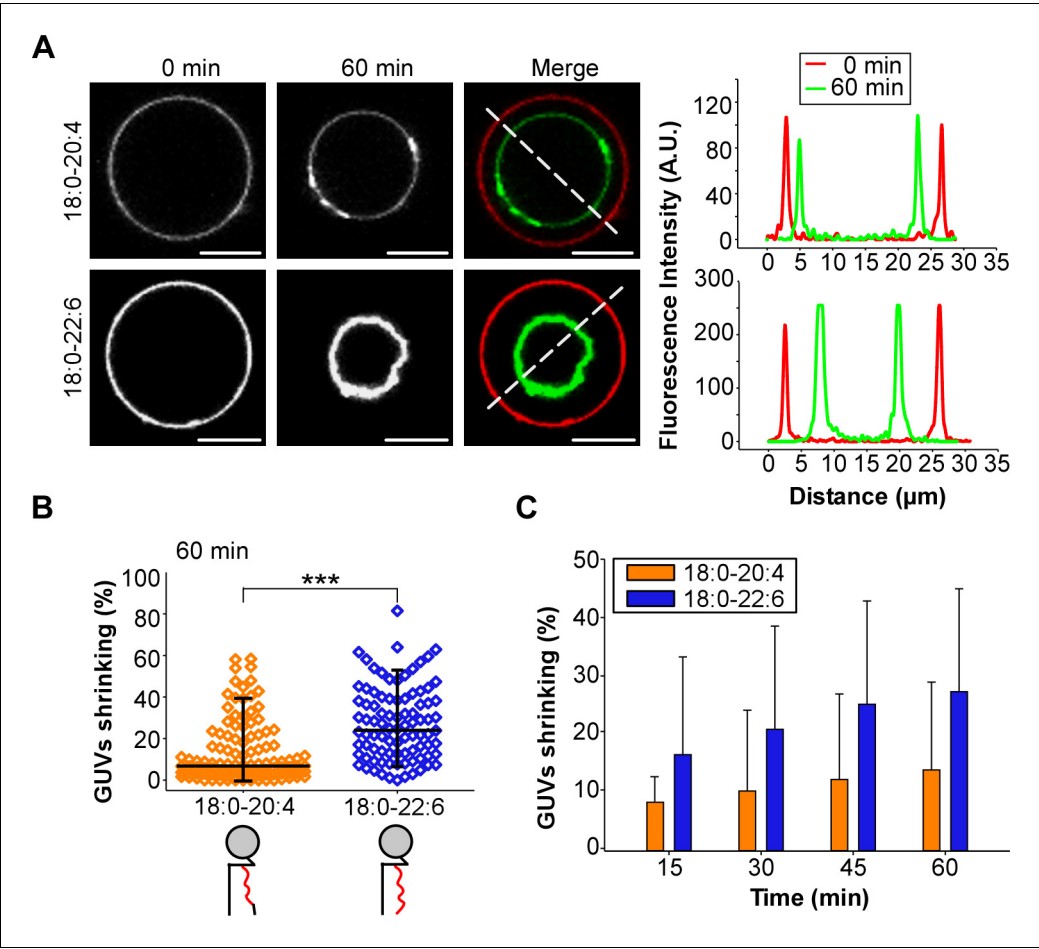

**Figure 5.** GUV shrinkage assay. GUVs containing 18:0-20:4 or 18:0-22:6 phospholipids were incubated with dynamin (0.5 µM) and endophilin (1 µM) in presence of GTP (500 µM) for 60 min at 37°C. (**A**) Colored images showing overlays of the GUVs at time 0 (red) and after 60 min incubation (green). Note the collapse of the GUV containing 18:0-22:6 after 60 min incubation with the protein-nucleotide mixture. (**B–C**) Quantification of the experiment shown in (**A**) showing the GUV shrinking distribution after 60 min and mean ±SD over the time. Data are obtained from ~100 GUVs from five independent preparations. All lipid compositions are showed in *Supplementary file 1*.

DOI: https://doi.org/10.7554/eLife.34394.011

The following figure supplement is available for figure 5:

**Figure supplement 1.** GUV shrinking assay with control monounsaturated membranes.
DOI: https://doi.org/10.7554/eLife.34394.012

the same curvature energy, these changes in L/R suggested inverse changes in membrane bending rigidity: 18:0-18:2, 18:0-20:4 and 18:0-22:6 membranes had relative values of $K_b$ equal to 50%, 35% and 30% of $K_b$ for 18:0-18:1 membranes, respectively.

During the time of the simulations (200 ns), we observed fission events for some tubes formed from 18:0-20:4 and 18:0-22:6 membranes but not from 18:0-18:1 or 18:0-18:2 membranes (*Figure 6B*, *Figure 6—figure supplement 1A* and *Video 1*). Although the number of simulations did not allow us to establish robust statistics, we noticed that the force threshold at which fission occurred was lower for 18:0-22:6 membranes than for 18:0-20:4 membranes (*Figure 6B* and *Figure 6—figure supplement 1B*). Moreover, fission occurred sooner for 18:0-22:6 tubes as compared to 18:0-20:4 tubes. Thus, the coarse-grained simulations agreed well with the experiments: the propensity of membranes to undergo deformation and fission correlates with the unsaturation level of the phospholipid *sn2* acyl chain.

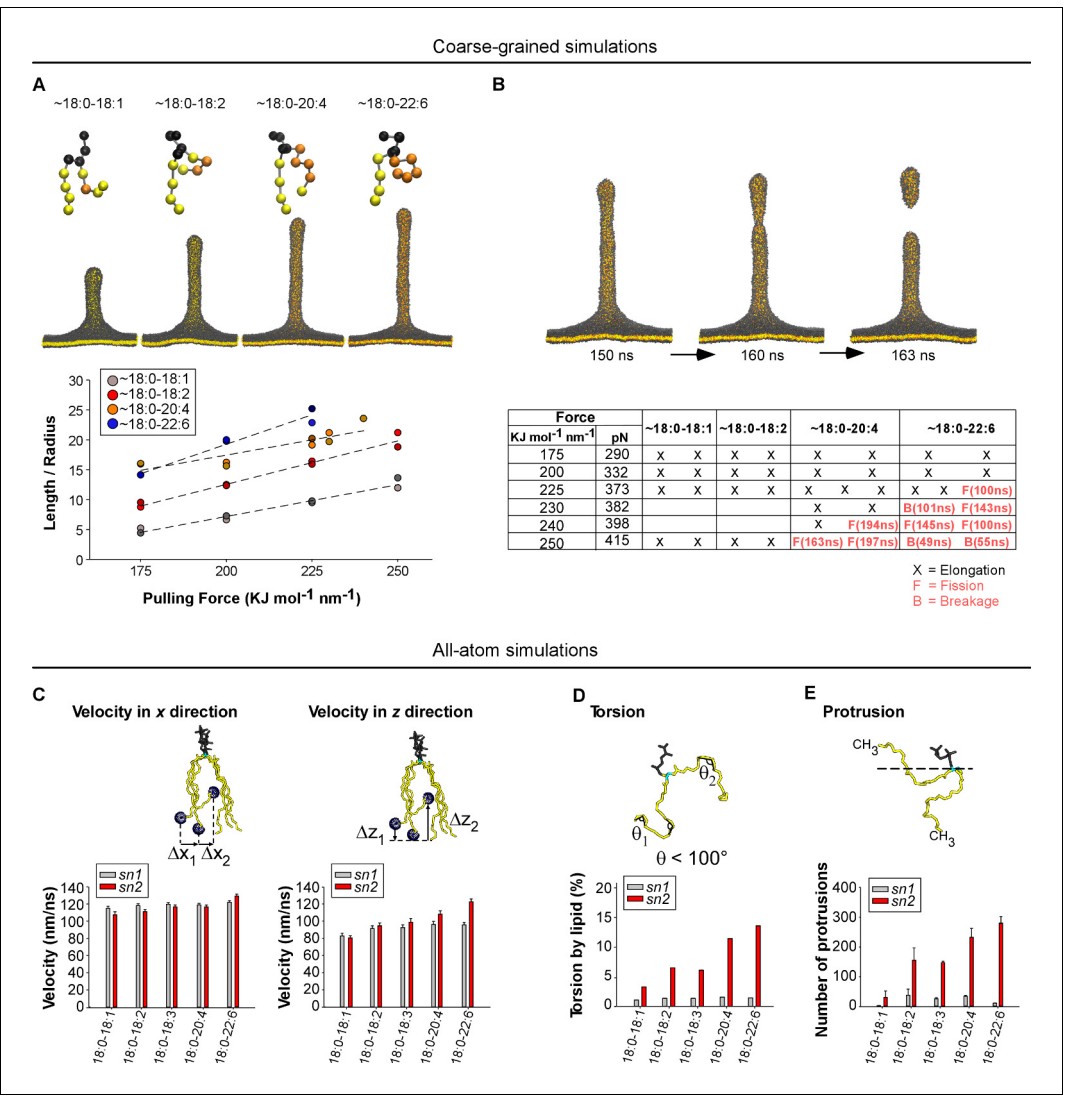

**Figure 6.** Molecular dynamics simulations. (**A**) Top: snapshots at t = 200 ns of tubes as pulled with a 200 KJ mol$^{-1}$ nm$^{-1}$ force from coarse-grained models of membranes with the indicated asymmetric polyunsaturated PC. Bottom: plot of $L/R$ as a function of the pulling force. The tube geometry was analyzed at t = 200 ns. (**B**) Top: snapshots of a tube pulling coarse-grained simulation (Force = 250 KJ mol$^{-1}$ nm$^{-1}$) obtained from a membrane with 18:0-20:4 PC and showing elongation and fission. See also **Video 1**. Bottom: summary of all simulations. Elongation, fission (**F**) and breakage events (**B**) are indicated. In contrast to fission, a breakage event corresponds to a rupture at the level of the lipids where the pulling force is applied. (**C–F**) Dynamics of the acyl chains of asymmetric polyunsaturated phospholipids from all-atom simulations. The analysis was performed on flat membrane patches with the same composition as that used in the experiments. (**C**) Velocity rate of the terminal CH$_3$ group of the acyl chain along either the membrane normal (z velocity) or in the membrane plane (x-y velocity). (**D**) Frequency of acyl chain torsions as defined as conformations for which the acyl chain displays an angle <100°. (**E**) Number of protrusions of the CH$_3$ group above the glycerol group of phospholipids. Color code for coarse-grained simulations: grey: lipid polar head and glycerol; orange: acyl chain regions with double carbon bonds; yellow: acyl chain regions with single carbon bonds. Color code for all-atom simulations: Grey: lipid polar head; cyan: glycerol; yellow: sn1 or sn2 acyl chains.

DOI: https://doi.org/10.7554/eLife.34394.013

The following figure supplements are available for figure 6:

**Figure supplement 1.** Coarse-grained molecular dynamics simulations of membrane deformation and fission.

DOI: https://doi.org/10.7554/eLife.34394.014

*Figure 6 continued on next page*

*Figure 6 continued*

**Figure supplement 2.** Distribution of deep and shallow lipid packing defects in all-atom simulations of membranes with the same composition as that used in the experiments of *Figure 1* and detailed in *Supplementary file 1*.

DOI: https://doi.org/10.7554/eLife.34394.015

**Figure supplement 3.** Comparison of phospholipids with a natural *sn1*-saturated-*sn2*-polyunsaturated profile and with a swapped *sn1*-polyunsaturated-*sn2*-saturated profile.

DOI: https://doi.org/10.7554/eLife.34394.016

For all-atom bilayers, we focused on parameters informative for the tendency of the phospholipid acyl chains to depart from the straight conformation. This tendency allows phospholipids to adopt different shapes and, consequently, to reduce the stress induced by membrane curvature (*Pinot et al., 2014*). We determined (i) the speed at which the terminal $CH_3$ group moves along the membrane normal (*z* velocity) *vs* membrane plane (*x* velocity); (ii) the frequency of FA torsions (when the acyl chain displays an angle <100°), (iii) the number of protrusions of the terminal $CH_3$ group above the glycerol during 100 ns; and (iv) the density of lipid packing defects, that is interfacial regions where aliphatic carbons are directly accessible to the solvent. *Figure 6C–E* and *Figure 6—figure supplement 2* show that whatever the parameter considered, the calculated value always increased with the polyunsaturation level of the *sn2* chain, with 22:6 clearly surpassing all other polyunsaturated FAs. In contrast, the behavior of the *sn1* 18:0 chain was relatively constant and appeared poorly dependent on the nature of the neighboring *sn2* chain. Altogether, these various analyses show that the main effect of having an *sn2* polyunsaturated chain in phospholipids is to increase the probability of fast movements of along the *z*-axis.

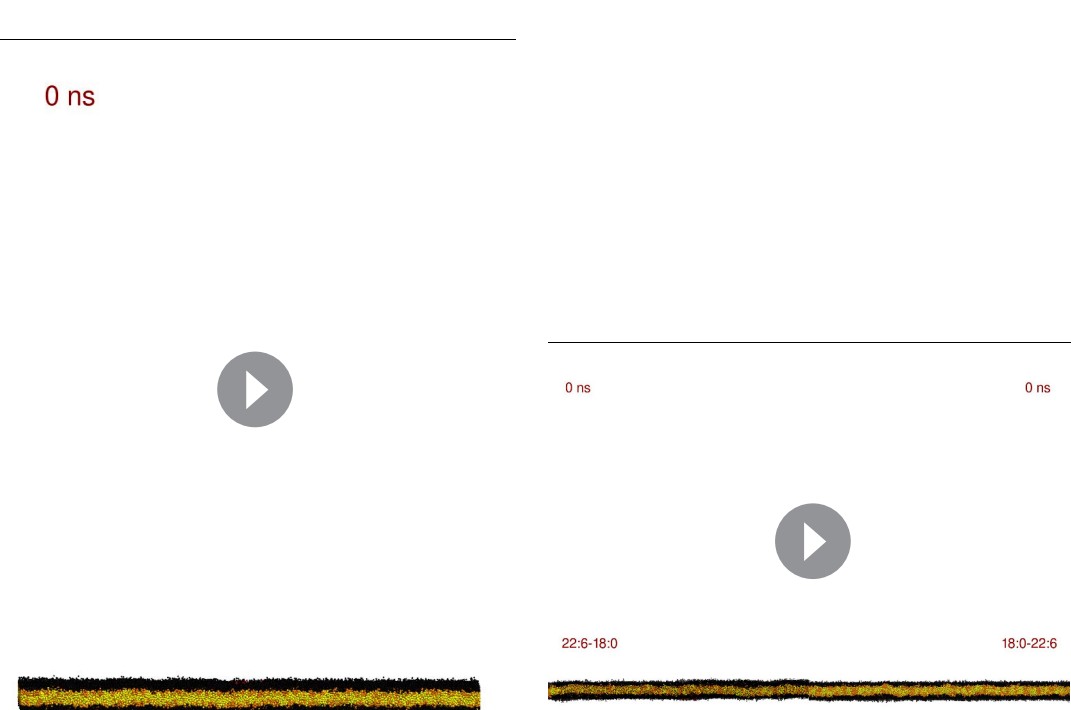

**Video 1.** Simulation of the formation of a tubule and its subsequent fission from a coarse-grained model of a membrane containing 18:0-20:4 phospholipids (see also *Figure 6B*) and submitted to a pulling force of 250 KJ mol$^{-1}$ nm$^{-1}$.

DOI: https://doi.org/10.7554/eLife.34394.017

**Video 2.** Simulation of the formation of a tubule from a coarse-grained bilayer containing 18:0-22:6 or 22:6-18:0 phospholipids (see also *Figure 6—figure supplement 3C*) and submitted to a pulling force of 200 KJ mol$^{-1}$ nm$^{-1}$.

DOI: https://doi.org/10.7554/eLife.34394.018

To determine whether the membrane features endowed by the polyunsaturated acyl chain depend on its esterification at position *sn2* as observed in natural lipids, we performed molecular dynamics simulations on phospholipid bilayers in which we swapped the *sn1* and *sn2* acyl chains (*Figure 6—figure supplement 3*). In all atom simulations, we observed that the rough features distinguishing the saturated and the polyunsaturated acyl chains remained after having permutated their position. These include *z* velocity, acyl chain torsions, and number of protrusions (*Figure 6—figure supplement 3A*). However, measurements of the acyl chain density across the bilayer indicated that acyl chain swapping modified the mean position of the saturated and polyunsaturated acyl chains across the bilayer (*Figure 6—figure supplement 3B*). This effect probably resulted from the tilted orientation of glycerol, which makes the *sn1* and *sn2* positions not equivalent in term of *z* coordinates. In natural phospholipids, the density profile of the *sn1* saturated FA showed a peak in the bilayer center whereas the *sn2* polyunsaturated FA showed a characteristic dip. This shift indicates that the *sn1* saturated FA tail invaded the central region of the bilayer left vacant by the *sn2* polyunsaturated FA, which goes up (*Eldho et al., 2003*). With swapped 22:6-18:0 phospholipids, this difference in density disappeared and the membrane appeared thinner than with natural 18:0-22:6 phospholipids (*Figure 6—figure supplement 3B*). Thus, the relative esterification position of the saturated and polyunsaturated FAs in natural lipids facilitates compensatory *z* movements where the polyunsaturated FA explores the interfacial region while the saturated FA explores the bilayer center. In coarse-grained simulations, the propensity of the membrane to undergo tubulation and fission increased with the level of phospholipid polyunsaturation (18:1-18:0 < 20:4-18:0 ≈ 22:6-18:0), that is the same trend as that observed on classical *sn1*-saturated-*sn2*-unsaturated membranes (*Figure 6—figure supplement 3C*). In addition, acyl chain swapping did not significantly modify membrane fission (*Figure 6—figure supplement 3C* and *Video 2*).

## Discussion

Although a wealth of information is available on the interactions between endocytic proteins and specific lipids (*Puchkov and Haucke, 2013*), the role of the hydrophobic membrane matrix has been poorly investigated. In vivo, manipulating the acyltransferases that are responsible for the large differences in the acyl chain profile of differentiated cells is challenging and is just starting to emerge (*Hashidate-Yoshida et al., 2015*; *Rong et al., 2015*). In vitro, purified lipids are generally available from disparate sources (e.g. egg PC, brain PS), implying different acyl chain profiles. Synthetic lipids provide the best alternative but the most affordable ones generally display symmetric acyl chain combinations. This is exemplified by DOPS (18:1-18:1 PS), which has allowed spectacular advances in our understanding of the structure of the dynamin spiral (*Chappie et al., 2011*), but is very rare in mammalian membranes (*Yabuuchi and O'Brien, 1968*). Overall, the membrane templates on which dynamin and its partners have been studied are generally ill defined in terms of acyl chain profiles. Our comprehensive analysis indicates that acyl chain asymmetry and acyl chain polyunsaturation have major effects on the mechanical activity of dynamin.

A few studies have established that polyunsaturated phospholipids considerably modify the properties of membranes (*Armstrong et al., 2003*; *Eldho et al., 2003*; *Garcia-Manyes et al., 2010*; *Huang, 2001*; *Olbrich et al., 2000*; *Rawicz et al., 2000*). For LUVs, a high dithionite permeability of membranes containing 18:3-18:3 phospholipids has been reported (*Armstrong et al., 2003*). For GUVs, micropipette manipulations indicate that the presence of at least one polyunsaturated acyl chain results in a drop of the membrane bending modulus whereas the presence of two polyunsaturated acyl chains causes a jump in water permeability (*Olbrich et al., 2000*; *Rawicz et al., 2000*). These pioneer studies were performed on membranes made of a single lipid (PC) with a limited combination of acyl chains and in the absence of mechanically active proteins. By using lipid mixtures covering a larger spectrum of acyl chain profiles and by including membrane shaping/fission proteins, our study underlines the importance of both phospholipid polyunsaturation and phospholipid acyl chain asymmetry in membrane mechanics.

Depending on its acyl chain profile, a membrane can be either very resistant or very permissive to dynamin-mediated membrane vesiculation despite harboring the proper repertoire of polar head groups for protein recruitment. However, these manipulations can also cause large changes in membrane permeability. Our analysis uncovers a narrow chemical window that allows phospholipid membranes to be both highly deformable and still impermeable to small solutes. Membranes with

asymmetric saturated-polyunsaturated phospholipids such as 18:0-20:4 or 18:0-22:6 phospholipids are much less leaky than membranes with symmetrical 20:4-20:4 or 22:6-22:6 phospholipids but can still be readily vesiculated by dynamin provided that BAR-domain proteins are present. Evidently, these features are advantageous for membranes such as synaptic membranes that undergo super-fast endocytosis (*Watanabe and Boucrot, 2017*). Furthermore, the fact that membranes with 18:0-22:6 phospholipids are systematically more permissive to the mechanical activity of dynamin and endophilin than membranes with 18:0-20:4 phospholipids is of interest given the importance of the omega-6/omega-3 ratio for health and notably for brain function.

The distinctive chemical feature of polyunsaturated acyl chains is the presence of saturated carbons ($CH_2$) sandwiched between two unsaturated ones (=CH-$CH_2$-CH=). Rotational freedom around these $CH_2$ groups is exceptionally high as compared to rotation around the $CH_2$ groups of monounsaturated or saturated acyl chains (*Feller et al., 2002*). Our MD simulations indicate that motions of the acyl chain along the normal of the membrane (*z* movements) increase in speed and in amplitude with the level acyl chain polyunsaturation (18:2 < 18:3 < 20:4 < 22:6). Such movements should allow phospholipids to readily adapt their conformation to membrane curvature (*Barelli and Antonny, 2016*; *Pinot et al., 2014*), hence explaining the gradual decrease in membrane bending rigidity. Concurrently, the presence of a neighboring saturated acyl chain should secure lipid packing and prevents the passage of small molecules. Whether this model also accounts for the facilitation of the fission step per se remains, however, difficult to assess. This step involves a change in membrane topology for which rare events such as protrusions of the terminal $CH_3$ groups could be decisive as they could nucleate bilayer merging or favor friction effects by proteins (*Simunovic et al., 2017*).

Other variations in the acyl chain content of mammalian phospholipids will deserve further investigations. First, we did not consider C22:5 acyl chains (omega-6 or omega-3), which are closer to C22:6 than C20:4 (*Eldho et al., 2003*). Although not abundant, C22:5 acyl chains are present in brain phospholipids (*Yabuuchi and O'Brien, 1968*). Second, we did not study the influence of the linkage between the acyl chains and glycerol. Plasmalogens, which form a large subclass of PE in the brain, harbor a *sn1* saturated acyl chain that is bound to the glycerol through an ether-vinyl bond. Interestingly, ether-vinyl phospholipids considerably decrease the permeability of model membranes to ions because these lipids pack more tightly than their ester counterparts (*Zeng et al., 1998*). The influence of plasmalogens on membrane flexibility and fission remains to be investigated. Last, we only partially addressed the bias observed in natural phospholipids, where saturated and polyunsaturated acyl chains are preferentially esterified on different positions of the glycerol backbone (*sn1* and *sn2*, respectively). Our simulations suggest that some general traits provided by the combination of one saturated acyl chain and one unsaturated acyl chain are preserved when the acyl chains are swapped between the *sn1* and *sn2* positions; notably the fact that membrane deformation and fission is facilitated by the level of polyunsaturation (18:1-18:0 < 20:4-18:0 ≈ 22:6-18:0). Testing this hypothesis by in vitro reconstitutions will require considerable efforts in lipid synthesis since swapped phospholipids are not commercially available.

The abilities to vesiculate and to act as selective barriers are two fundamental properties of cellular membranes. Without membrane vesiculation, a cell cannot divide; without selective permeability, it cannot control the concentration of its nutrients. Experiments aimed at mimicking the emergence of primitive membranes have illuminated how these properties need to be finally balanced. Single chain amphiphilic molecules (e.g. fatty acids), the most plausible building blocks for primitive membranes, can self-assemble into bilayers, which spontaneously vesiculate (*Bruckner et al., 2009*). However, these bilayers are very leaky to even large solutes (in the $10^3$ Da range). Later, the shift from single chain to dual chain lipids has probably allowed primitive cells to reduce the general permeability of their membrane, thereby imposing an evolutionary pressure for the emergence of specialized transporters (*Budin and Szostak, 2011*). The experiments presented here suggest that phospholipids with one saturated and one polyunsaturated acyl chain, which are absent in many eukaryotes (e.g. yeast) but abundant in some highly differentiated cells (e.g. neurons, photoreceptors, sperm) provide a solution to an early dilemma in evolution: finding the right balance between efficient membrane vesiculation without loss in membrane permeability. Moreover, the fact that saturated-DHA (omega-3) phospholipids are systematically better for membrane vesiculation than many other saturated-polyunsaturated phospholipids, including saturated-arachidonate (omega-6), is informative considering the importance of the omega-6/omega-3 ratio for health.

# Materials and methods

### Key resources table

| Reagent type (species) or resource | Designation | Source or reference | Identifiers | Additional information |
|---|---|---|---|---|
| Phosphatidylcholine | 1-stearoyl-2-oleoyl-sn-glycero-3-phosphocholine | Avanti Polar Lipids | Ref 18:0-18:1 PC \| 850467 | |
| Phosphatidylcholine | 1-stearoyl-2-linoleoyl-sn-glycero-3-phosphocholine | Avanti Polar Lipids | Ref 18:0-18:2 PC \| 850468 | |
| Phosphatidylcholine | 1-stearoyl-2-linolenoyl-sn-glycero-3-phosphocholine | Avanti Polar Lipids | | 18:0-18:3 PC Custom |
| Phosphatidylcholine | 1-stearoyl-2-arachidonoyl-sn-glycero-3-phosphocholine | Avanti Polar Lipids | Ref 18:0-20:4 PC \| 850469 | |
| Phosphatidylcholine | 1-stearoyl-2-docosahexaenoyl-sn-glycero-3-phosphocholine | Avanti Polar Lipids | Ref 18:0-22:6 PC \| 850472 | |
| Phosphatidylcholine | 1,2-dimyristoyl-sn-glycero-3-phosphocholine | Avanti Polar Lipids | Ref 14:0 PC (DMPC) \| 850345 | |
| Phosphatidylcholine | 1,2-dioleoyl-sn-glycero-3-phosphocholine | Avanti Polar Lipids | Ref 18:1 (Δ9-Cis) PC (DOPC) \| 850375 | |
| Phosphatidylcholine | 1,2-dilinoleoyl-sn-glycero-3-phosphocholine | Avanti Polar Lipids | Ref 18:2 (Cis) PC (DLPC) \| 850385 | |
| Phosphatidylcholine | 1,2-diarachidonoyl-sn-glycero-3-phosphocholine | Avanti Polar Lipids | 20:4 (Cis) PC \| 850397 | |
| Phosphatidylcholine | 1,2-didocosahexaenoyl-sn-glycero-3-phosphocholine | Avanti Polar Lipids | 22:6 (Cis) PC \| 850400 | |
| Phosphoethanolamine | 1-stearoyl-2-oleoyl-sn-glycero-3-phosphoethanolamine | Avanti Polar Lipids | 18:0-18:1 PE \| 850758 | |
| Phosphoethanolamine | 1-stearoyl-2-linoleoyl-sn-glycero-3-phosphoethanolamine | Avanti Polar Lipids | 18:0-18:2 PE \| 850802 | |
| Phosphoethanolamine | 1-stearoyl-2-linolenoyl-sn-glycero-3-phosphoethanolamine | Avanti Polar Lipids | | 18:0-18:3 PE Custom |
| Phosphoethanolamine | 1-stearoyl-2-arachidonoyl-sn-glycero-3-phosphoethanolamine | Avanti Polar Lipids | Ref 18:0-20:4 PE \| 850804 | |
| Phosphoethanolamine | 1-stearoyl-2-docosahexaenoyl-sn-glycero-3-phosphoethanolamine | Avanti Polar Lipids | Ref 18:0-22:6 PE \| 850806 | |
| Phosphoethanolamine | 1,2-dimyristoyl-sn-glycero-3-phosphoethanolamine | Avanti Polar Lipids | Ref 14:0 PE \| 850745 | |
| Phosphoethanolamine | 1,2-dioleoyl-sn-glycero-3-phosphoethanolamine | Avanti Polar Lipids | Ref 18:1 (Δ9-Cis) PE (DOPE) \| 850725 | |
| Phosphoethanolamine | 1,2-dilinoleoyl-sn-glycero-3-phosphoethanolamine | Avanti Polar Lipids | Ref 18:2 PE \| 850755 | |
| Phosphoethanolamine | 1,2-diarachidonoyl-sn-glycero-3-phosphoethanolamine | Avanti Polar Lipids | Ref 20:4 PE \| 850800 | |
| Phosphoethanolamine | 1,2-didocosahexaenoyl-sn-glycero-3-phosphoethanolamine | Avanti Polar Lipids | Ref 22:6 PE \| 850797 | |
| Phosphatidylserine | 1-stearoyl-2-oleoyl-sn-glycero-3-phospho-L-serine | Avanti Polar Lipids | Ref 18:0-18:1 PS \| 840039 | |
| Phosphatidylserine | 1-stearoyl-2-linoleoyl-sn-glycero-3-phospho-L-serine | Avanti Polar Lipids | Ref 18:0-18:2 PS \| 840063 | |
| Phosphatidylserine | 1-stearoyl-2-linolenoyl-sn-glycero-3-phospho-L-serine | Avanti Polar Lipids | | 18:0-18:3 PS Custom |
| Phosphatidylserine | 1-stearoyl-2-arachidonoyl-sn-glycero-3-phospho-L-serine | Avanti Polar Lipids | Ref 18:0-20:4 PS \| 840064 | |
| Phosphatidylserine | 1-stearoyl-2-docosahexaenoyl-sn-glycero-3-phospho-L-serine | Avanti Polar Lipids | Ref 18:0-22:6 PS \| 840065 | |

*Continued on next page*

*Continued*

| Reagent type (species) or resource | Designation | Source or reference | Identifiers | Additional information |
|---|---|---|---|---|
| Phosphatidylserine | 1,2-dimyristoyl-sn-glycero-3-phospho-L-serine | Avanti Polar Lipids | Ref 14:0 PS | 840033 | |
| Phosphatidylserine | 1,2-dioleoyl-sn-glycero-3-phospho-L-serine | Avanti Polar Lipids | Ref 18:1 PS (DOPS) | 840035 | |
| Phosphatidylserine | 1,2-dilinoleoyl-sn-glycero-3-phospho-L-serine | Avanti Polar Lipids | Ref 18:2 PS | 840040 | |
| Phosphatidylserine | 1,2-diarachidonoyl-sn-glycero-3-phospho-L-serine | Avanti Polar Lipids | Ref 20:4 PS | 840066 | |
| Phosphatidylserine | 1,2-didocosahexaenoyl-sn-glycero-3-phospho-L-serine | Avanti Polar Lipids | Ref 22:6 PS | 840067 | |

## Protein purification and labelling

Proteins were purified as described (*Pinot et al., 2014*; *Stowell et al., 1999*). Dynamin was purified from rat brain using a recombinant amphiphysin-2 SH3 domain as an affinity ligand. Brain extracts were incubated with 10 mg ml$^{-1}$ glutathione-S-transferase-tagged amphiphysin-2 SH3 domain on glutathione–agarose beads at 4°C. After extensive washing of the matrix in buffer A (100 mM NaCl, 20 mM HEPES, pH 7.3, 1 mM dithiothreitol (DTT)), dynamin was eluted in 3 ml buffer B (1.2 M NaCl, 20 mM HEPES, pH 6.5, 1 mM DTT), and dialysed overnight into 200 mM NaCl, 20 mM HEPES, 20% glycerol. Full-length mouse endophilin A1 in pGEX-6p1 (gift of A. Schmidt) was expressed in E coli for 3 hr at 37°C after induction with 1 mM IPTG. Cells were lysed in 50 mM Tris pH 7.4, 150 mM NaCl using a French press in the presence of antiproteases and spun at 40,000 rpm for 30 min at 4°C. The supernatant was incubated with glutathione-Sepharose 4B beads followed by extensive washes in lysis buffer. PreScission protease was directly added to the beads at 4°C overnight under gentle agitation to cleave the fusion protein. Endophilin was recovered in supernatant and further purified on a Superdex 200 column in 20 mM Tris pH 7.4, 300 mM KCl, 5 mM imidazole, 1 mM DTT.

## Preparation of liposomes

Lipids were purchased from Avanti Polar Lipids as chloroform solutions (see Key resources table). These included the following species of phosphatidylcholine (PC), phosphatidylethanolamine (PE) and phosphatidylserine (PS): 14:0-14:0, 18:0-18:1, 18:1-18:1, 18:0-18:2, 18:2-18:2, 18:0-18:3, 18:0-20:4, 20:4-20:4, 18:0-22:6 and 22:6-22:6. Note that 18:0-18:3 phospholipid species were custom-made lipids from Avanti. Phosphatidylinositol (4,5)bisphosphate (PIP$_2$) was from natural source (brain). Submicrometer liposomes used for biochemical experiments and for electron microscopy were prepared by extrusion. A lipid film containing phospholipids and cholesterol at the desired molar ratio (see Table S1-3 in *supplementary file 1*) was formed in a rotary evaporator and hydrated at a final lipid concentration of 1 mM in a freshly degassed HK buffer (50 mM Hepes pH 7.2, 120 mM K Acetate) supplemented with 1 mM DTT. The suspension was submitted to five cycles of freezing and thawing and stored at −20°C under argon to avoid lipid oxidation. Calibrated liposomes were obtained by extrusion through 400 or 100 nm polycarbonate filters using a hand extruder (Avanti Polar Lipids). The size distribution of the liposomes was determined by dynamic light scattering at a final concentration of 0.1 mM lipids in HK buffer. All liposome suspensions were used within 1–2 days after extrusion. Special care was taken to minimize lipid oxidation by using fleshly degassed buffer (supplemented with 1 mM DTT) and by storing the liposome suspensions under argon.

## Preparation of GUVs

Giant unilamellar vesicles were generated by electroformation as described (*Pinot et al., 2014*) with the following modifications. Lipid mixtures (0.5 mg/ml; see Table S4 in *supplementary file 1*) in chloroform were deposited on indium tin oxide coated glass slides at 50°C to prevent lipid de-mixing and dried under vacuum for 1 hr to remove all solvents. After this step, sucrose 250 mM osmotically equilibrated with buffers was added to the chamber. GUVs were electroformed (*Angelova et al., 1992*) with Vesicle Prep Pro (Nanion Technologies GmbH, Munich, Germany), applying an AC electric field with 3 V and 5 Hz for 218 min at 37°C.

## GTPase assay

GTP hydrolysis in dynamin was measured using a colorimetric assay (*Leonard et al., 2005*). The sample (60 µl) initially contained 400 nm extruded liposomes (0.1 mM) of defined composition (see *Supplementary file 1*) in HK buffer supplemented with 2.5 mM $MgCl_2$, 1 mM DTT and 500 µM GTP. Just before measurement, endophilin (0.6 µM) was added and the reaction was initiated by the addition and mixing of 0.3 µM dynamin. At the indicated times (15, 45, 75, 120, 180, 240 and 360 s), aliquots (7.5 µl) were withdrawn and immediately mixed with a drop of EDTA (5 µl, 250 mM) in a 96 well plate. At the end of the experiment, 150 µl of a malachite green stock solution was added to each well and the absorbance at 650 nm was measured using a microplate reader and compared to that of a standard curve of phosphate (0–200 µM) in order to determine the concentration of GTP hydrolyzed by dynamin.

## Push-pull pyrene (PA) fluorescence on LUVs

Fluorescence spectra of the PA probe with liposomes was performed as described (*Niko et al., 2016*). The sample (600 µl) initially contained 0.1 mM extruded liposomes (100 nm) of defined composition (see *Supplementary file 1*). After 5 min incubation of the liposomes solution with 1 µM PA probe at 37°C, a fluorescence emission spectrum (450–700 nm; bandwidth 1 nm) was recorded upon excitation at 430 nm (bandwidth 5 nm). All spectra were corrected for the corresponding blank (suspension of liposomes without the probe).

## Dithionite-mediated NBD quenching assay

The extent of dithionite quenching of the NBD-labeled PE was performed as described (*Angeletti and Nichols, 1998*). Briefly, the sample (600 µl) that initially contained 400 nm extruded liposomes (0.1 mM) of defined composition (see *Supplementary file 1*) were let equilibrating in HK buffer 5 min at 37°C with 600 rpm stirring. After 30 s of fluorescence measurements (excitation 505 nm, bandwidth 1 nm; emission 540 nm, bandwidth 10 nm), NBD quenching was started by adding 10 mM dithionite and the reaction was followed during 5 min at 37°C with 600 rpm stirring. The percentage of NBD quenching was calculated by the equation: Quenching NBD (%) = $(F_i - F_0) / (F_T - F_0)$ x 100 where $F_0$ corresponds to the fluorescence of the vesicles at time 0–30 s; $F_i$ is the fluorescence after a certain period of incubation with dithionite, and $F_T$ is the maximum quenching that corresponds to the fluorescence value obtained after addition of 0.1% Triton X-100.

## Electron microscopy

Mixtures containing liposomes, dynamin, endophilin and nucleotides were prepared in HK buffer supplemented with 2.5 mM $MgCl_2$ and 1 mM DTT (final volume 50 µl). For the tubulation experiments in presence of the non-hydrolyzable analog GTPγS, vesicles were incubated for 5 or 15 min at room temperature. For the fission experiments in presence of GTP, vesicles were incubated for 30 min at room temperature. Thereafter, an EM grid was put on the protein-liposome mixture for 5 min, rinsed with a droplet of 100 mM Hepes (pH 7.0) for 1 min, and then stained with 1% uranyl acetate. The grid was observed in a JEOL JEM1400 transmission electron microscope equipped with a MORADA SIS camera. To determine the size distribution of the liposomes or of the protein-liposome profiles, 500 to 1000 profiles for each condition and from three independent experiments were analyzed using the ellipse tool of the NIH Image J software. The apparent radius was calculated as $R= (A/\pi)^{1/2}$ where A is the apparent area of the profile. All experiments were performed with 0.5 µM dynamin, 1 µM endophilin, 500 µM nucleotide and 0.1 mM lipids.

## GUV permeability and size assay

GUV permeability was studied in 18:0-22:6 and 22:6-22:6 liposomes using a previously developed assay with some modifications (*Jiménez-Rojo et al., 2014*). After GUVs stabilization soluble Alexa488 was externally added to follow the entrance of the probe over time. After 15 min incubation, vesicles were imaged by confocal microscopy and permeability was quantified using the following equation: Permeability (%)=$I_{in}/I_{ex}$ x 100 where $I_{in}$ is the average of the fluorescence inside the individual GUV and $I_{ex}$ is the average of external fluorescence of the probe in solution.

Membrane fission induced by dynamin and endophilin in the presence of GTP was followed indirectly by monitoring the size of GUVs over time since the vesicles produced by the proteins are too

small to be optically resolved. We used a previously developed assay with some modifications (*Meinecke et al., 2013*). After GUVs stabilization, dynamin, endophilin and GTP were added and incubated for 1 hr at 37°C before and imaging by confocal microscopy. All experiments were performed with 0.5 µM dynamin, 1 µM endophilin, 500 µM nucleotide and 0.1 mM lipids. Shrinking percentage was calculated by the equation: Shrinking (%)=100 x $A_0/A_i$ where $A_0$ is the vesicles area at time 0 and $A_i$ is vesicles area after a defined period of incubation.

## Molecular dynamics

All-atom simulations were performed with GROMACS 5 (*Abraham et al., 2015*) software and CHARMM36 (*Klauda et al., 2010*) force field. The various systems were built with the Charmm-Gui tool (*Lee et al., 2016*) with 33% Cholesterol, 1% PI(4,5)$P_2$, and with 30% PS, 20% PE, and 16% PC, harboring defined acyl chains (18:0-18:1, 18:0-18:2, 18:0-18:3, 18:0-20:4, 18:0-22:6, 14:0-14:0, 18:2-18:2, 20:4-20:4, 22:6-22:6). Lipids not present in the Charmm-Gui database (18:0-18:2, 18:0-18:3, 18:2-18:2 and 22:6-22:6 and swapped lipids) were built by adding unsaturations to related lipids (i.e. 18:0-18:1, 18:1-18:1 or 22:1-22:1). Note that one of this lipid (18:0-18:2) is now present in the database and has the same topology as the one used here. The bilayers contained 2 × 144 phospholipids with counter ions to neutralize the system and with 120 mM NaCl.

The simulation parameters were those of Charmm-Gui under semi isotropic conditions within the NPT ensemble: *x* and *y* directions were coupled, whereas *z* direction was independent. Periodic boundaries applied to all directions. We first equilibrated the membranes for 200 ps using the standard Charmm-Gui six-step process during which constraints on lipids were gradually released. Next, an additional equilibration step was performed to equilibrate the TIP3P model of water. All simulations were equilibrated using the Berendsen thermostat and barostat at 303 K and 1 bar, respectively, except for 14:0-14:0 bilayers, which were equilibrated at 310 K. Lipids and water+ions were coupled separately.

Production runs were performed with the V-rescale thermostat at 303 K except for 14:0-14:0 bilayers (310 K). The Parrinello-Rahman thermostat was used to stabilize the pressure at 1 bar with a time constant of 5 ps and a compressibility of 4.5 × $10^{-5}$ $bar^{-1}$ (*Parrinello and Rahman, 1981*). Again, lipids and water+ions were coupled separately. The time step was set at 2 fs. Bond lengths were constrained using the P-LINCS algorithm (*Hess, 2008*). Cutoff was fixed at 1.2 nm for the Lennard-Jones and electrostatic interactions. The smooth particle-mesh was used to evaluate the electrostatic interactions. Frames were saved every 10 ps.

Trajectory analyses were performed from 400 ns simulations from which we discarded the first 100 ns in order to rule out processes that are not at equilibrium. The remaining 300 ns trajectory was divided in 3 blocks of 100 ns to determine the standard deviation. Frames were analyzed every 100 ps except for the velocity and permeability analysis for which we used 10 ps frames.

Coarse-grained simulations were performed with GROMACS 4.5 (*Hess et al., 2008*) using the Martini force field (*Wassenaar et al., 2015*). The systems were built with the Charmm-Gui tool adapted to coarse-grained simulations (*Qi et al., 2015*). In all simulations, we varied the acyl chains composition while keeping the PC polar head constant. We used four lipids to approximate the asymmetric lipids 18:0-18:1, 18:0-18:2, 18:0-20:4, and 18:0-22:6. Note that the coarse-grained simplification does not distinguish C16:0 from C18:0, C20:4 from C20:5, and C22:6 from C22:5. We built coarse-grained models of swapped phospholipids from natural phospholipids having acyl chains of the same length. The systems contained 18 000 lipids and were solvated with a 100 nm thick layer of water.

The sytems were equilibrated with the standard Charmm-Gui six-step process. Production runs were performed with the V-rescale thermostat at 303K. The Berendsen barostat was used to stabilize the pressure at 1 bar with a time constant of 4ps and compressibility of 5 × $10^{-5}$ $bar^{-1}$ (*Berendsen et al., 1984*). The different membranes were simulated under a semi-isotropic condition and the periodic boundaries were applied in all directions. Lipids and water/ions were coupled separately. The time step was fixed at 20 fs and the cutoff for the Lennard-Jones and electrostatic interactions was set at 1.2 nm. The smooth particle-mesh was used to evaluate the electrostatic interactions.

To simulate membrane deformation and fission, we applied a force perpendicular to the initially flat bilayer (*Baoukina et al., 2012*). The force (from 175 to 250 KJ $mol^{-1}$ $nm^{-1}$) was applied to the center of mass of a lipid patch of radius = 3 nm, in which lipids were restrained in the lateral (*x,y*)

directions. The simulations were performed for 200 ns and were repeated two to three times under most conditions.

For further information on all molecular dynamics simulations, refer to the Gromacs mdp files (*supplementary files 2* and *3*).

## Simulation analysis

To evaluate membrane permeability, we counted the number of water molecule(s) that have visited the center of membrane (corresponding to 65% of the thickness) during 100 ns time frames. These water molecules were separated in two classes: those that fully crossed the bilayer and those that entered the hydrophobic region and then exited from the same side. Results were normalized to the total number of water molecules. The velocity rate of the terminal methyl group of the acyl chains was calculated from the sum of distances traveled by each methyl group in the $x$ or in the $z$ direction every 10 ps. For protrusions, we calculated the number of events during 100 ns blocks where the $CH_3$ terminal group of the acyl chain reached a $z$ position above the central carbon of glycerol from the same lipid. An acyl chain torsion corresponds to an angle below 100° between carbons that have relative positions of n-2, n and n+2 along the acyl chain. Packing defect analysis was performed as previously described (*Vamparys et al., 2013*). This membrane scanning procedure allows the detection of aliphatic atoms that are directly accessible to the solvent and that are either <1 Å (shallow defect) or >1 Å (deep defect) below the nearest glycerol.

## Acknowledgements

We thank Andrey Klymchenko for the PA probe, Wen-Ting Lo and Volker Haucke for the SNX9 protein, Guillaume Drin for help with GUVs preparation and Alenka Copic for comments on the manuscript. We are very grateful to Hugues Chap, Michel Lagarde, and Gérard Lambeau for their insights into the history of acyl chain asymmetry of phospholipids. This work was supported in part by an ERC grant (268 888) and is currently supported by the Agence Nationale de la Recherche (ANR-11-LABX-0028–01) and the HPC resources of CINES under the allocations 2016-c2016077362 and 2017-A0020707362 made by GENCI. MM is supported by a postdoctoral fellowship from the Basque Government.

## Additional information

### Funding

| Funder | Grant reference number | Author |
| --- | --- | --- |
| Agence Nationale de la Recherche | ANR-11-LABX-0028-01 | Bruno Antonny |
| European Commission | ERC advanced grant 268888 | Bruno Antonny |
| Eusko Jaurlaritza | Post-doctoral fellowship | Marco M Manni |

The funders had no role in study design, data collection and interpretation, or the decision to submit the work for publication.

### Author contributions

Marco M Manni, Conceptualization, Data curation, Formal analysis, Investigation, Methodology, Writing—original draft; Marion L Tiberti, Conceptualization, Data curation, Formal analysis, Investigation, Methodology; Sophie Pagnotta, Investigation, Methodology; Hélène Barelli, Supervision; Romain Gautier, Conceptualization, Software, Formal analysis, Supervision, Investigation, Methodology; Bruno Antonny, Conceptualization, Formal analysis, Funding acquisition, Investigation, Writing—original draft, Project administration, Writing—review and editing

### Author ORCIDs

Bruno Antonny (iD) http://orcid.org/0000-0002-9166-8668

**Decision letter and Author response**
Decision letter https://doi.org/10.7554/eLife.34394.024
Author response https://doi.org/10.7554/eLife.34394.025

---

## Additional files

### Supplementary files

• Supplementary file 1. Tables of the lipid composition of the various bilayers used in this study.
DOI: https://doi.org/10.7554/eLife.34394.019

• Supplementary file 2. Mdp input file of the all atom simulations.
DOI: https://doi.org/10.7554/eLife.34394.020

• Supplementary file 3. Mdp input file of the coarse-grained simulations
DOI: https://doi.org/10.7554/eLife.34394.021

• Transparent reporting form
DOI: https://doi.org/10.7554/eLife.34394.022

---

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
