## [Decision Letter]

Thank you for submitting your article "Saturated-docosahexaenoic phospholipids have an optimal acyl chain combination for membrane vesiculation without leakage" for consideration by *eLife*. Your article has been favorably evaluated by Vivek Malhotra (Senior Editor) and three reviewers, one of whom, Patricia Bassereau (Reviewer #1), is a member of our Board of Reviewing Editors.

The reviewers have discussed the reviews with one another and the Reviewing Editor has drafted this decision to help you prepare a revised submission.

Summary:

First, I would like to mention that both referees and myself appreciated your manuscript very much and the new perspectives it opens on the role of lipids in endo-exocytosis as well as in evolution. Indeed, this paper addresses an interesting issue in membrane biology – how can cell membranes be so flexible that they are able to vesiculate and divide without compromising their impermeability.

Practically, the manuscript examines the effect of polyunsaturated fatty acyl chains in lipid membranes. A range of techniques were used to study the effects of varying the number and location of double bonds, the chain length, and the location of the polyunsaturated chains (*sn1* vs. *sn2*), in particular a dynamin remodeling assay, membrane permeability probes, and molecular dynamics simulations. The study is a tour de force and complements the biochemical data with molecular dynamics analysis. The authors find that phospholipids with polyunsaturated chain at position *sn2* strike an ideal compromise between membrane deformability and permeability. Their results nicely explain why natural membranes never contain symmetric PULs and provide a rationale for the relative abundance of polyunsaturated lipids, e.g., in neurons.

Essential revisions:

Your manuscript will be suitable for publication in *eLife* provided you can address the following rather straightforward issues.

1) What happens when the polyunsaturated chain and the saturated one are swapped? So far, you only considered the polyunsaturated chain is at position *sn2* and not *sn1*. This could be discussed further. We also leave you with the option of addressing this with simulations. This would nicely complement the work.

2) Please provide more details on the MD simulations: the MD parameter input files (.mdp for GROMACS) could be included in the supplementary information, both for the all-atom and the coarse-grained simulations.

3) The results of the dithionite quenching assay should be discussed in more detail (Figure 2C). It is surprising that the system rapidly plateaus at a value lower than 100% if the membrane is leaky to dithionite and not at 100%.

---

## [Author Response]

Essential revisions:Your manuscript will be suitable for publication in eLife provided you can address the following rather straightforward issues.1) What happens when the polyunsaturated chain and the saturated one are swapped? So far, you only considered the polyunsaturated chain is at position sn2 and not sn1. This could be discussed further. We also leave you with the option of addressing this with simulations. This would nicely complement the work.

This is a highly relevant point, which would require two approaches to be fully addressed:

Experiments. We contacted our lipid provider (Avanti) because these swapped lipids are not commercially available. For custom synthesis, their proposal is the following: “The cost for one complete series, like 20:4-18:0 PC,PS and PE (50 mg each, minimum order), is $12,000 USD. It will take ~10 weeks to ship the material to you.”

Because as we need to analyze at least three series to get a robust conclusion (18:1, 20:4 and 22:6), the cost (36,000 $) will be too high for our budget. In addition, we will require more time to perform the biochemical measurements after having received the lipids (at best 3 or 4 additional months), which will seriously postpone the publication.

Simulations. Following your advice, we conducted all atom and coarse-grained simulations (new Figure 6—figure supplement 3). In all atom simulations (i.e. on small flat membrane patches), we observed that the swapped lipids behave quite similarly as the normal lipids (e.g. C20:4-18:0 versus 18:0-20:4). For example, the differences in speed along the z axis between the polyunsaturated acyl chain and the saturated acyl chain remained, regardless of their esterification position. In coarse-grained simulations, we also observed that membranes made of *sn1*-polyunsaturated-*sn2*-saturated PLs (e.g. 22:6-18:0 or 20:4-18:0) undergo fission more readily than the control (18:1-18:0) membranes, i.e. the same trend as that observed on classical *sn1*-saturated-*sn2*-unsaturated membranes. Surprisingly, however, acyl chain swapping facilitates membrane fission (22:618:0 > 18:0-22:6). This difference might arise from the fact that the density profile of the acyl chains in the swapped lipids is less well balanced than in the natural lipids, which might make the bilayer less stable. However, we cannot make firm statements because these observations derive only from simulations.

In the revised text, we use the new simulations to enrich the Discussion and suggest new lines for the future. On a related issue, we also mention in the same paragraph that it will be worth studying other factors related to acyl chain asymmetry. In the brain, a large fraction of PE shows a vinyl-ether linkage at the sn1 position instead of an ester linkage. These species, defined as plasmalogens, pack differently as compared to classical phospholipids, but have never been studied in the context of membrane deformation.

2) Please provide more details on the MD simulations: the MD parameter input files (.mdp for GROMACS) could be included in the supplementary information, both for the all-atom and the coarse-grained simulations.

We have expanded the Materials and methods section as well as the supplementary information to provide much more details about the MD simulations. The MD parameter input files (.mdp) are now available as supplementary files.

3) The results of the dithionite quenching assay should be discussed in more detail (Figure 2C). It is surprising that the system rapidly plateaus at a value lower than 100% if the membrane is leaky to dithionite and not at 100%.

Thank you for raising this point. We agree that the NBD quenching assay should be discussed in more details. This process depends on several factors:

- Liposome size. On small liposomes, conical lipids like C18:1-PE-NBD tend to be enriched in the inner leaflet because their intrinsic shape fits better with a convex surface than a concave one. Consequently, the amplitude of the fast phase (quenching of PE-NBD present on the outer leaflet) is < 50% (Kamal et al. (2009). This factor is generally modest and should not play a role in our experiments because we use large liposomes (extrusion 0.4 µm). Note that the choice of large liposomes was motivated by the fact that we wanted to study membrane permeability with the same liposomes as that used in the dynamin GTPase experiments. In fact, the liposomes used for the NBD quenching experiments (Figure 2C) were also used in the GTPase experiments (Figure 1B). As such, we can fairly compare the two measurements.

- Number of bilayers. Liposomes extruded through large pores (here 0.4 µm) are not perfectly unilamellar in contrast to those extruded through smaller pores (< 50 nm). Because the encapsulated bilayers are not directly accessible to dithionite: this effect leads to quenching amplitude < 50% Kamal et al. (2009). This effect probably applies here: if we extrapolate at time zero the kinetics of PE-NBD quenching for the liposomes with mixed acyl chain phospholipids, which are clearly poorly permeable (central panel of Figure 2C), we obtain values slightly below 50% (e.g. 18:0-18:1). For highly permeable liposomes (e.g. 22:6-22:6), the uncomplete quenching at time 350 s, probably arises from the fact that it takes more time for dithionite to pass through N bilayers than to one, hence this apparent uncomplete quenching effect. Note, however, that at time 350 s, the signal is not perfectly flat and that quenching still increases slowly.

- Liposomes charge (See Zeng et al., 1998. The authors of this paper show that negatively charged lipid can counteract membrane permeability; changing the polar head (PC > PS) while keeping the acyl chain constant induces a strong decrease in membrane permeability to ion. This factor is certainly at play here because our liposomes contain a fraction of charged lipids. However, all liposome preparations have the same composition in term of polar head; thus, this effect should be rather constant.

- Lipid flip-flop. This factor has been thoroughly analyzed by Zeng et al., 1998. The authors use a protocol in which external PE-NBD is first quenched by dithionite for a few seconds, then the excess of dithionite is immediately separated by a gel filtration step. The liposomes are left in buffer for a given amount of time. Dithionite is then added again to probe the fraction of PE-NBD that has flipped flop during the incubation. These experiments show a large effect of lipid polyunsaturation and symmetry vs. asymmetry but also that flip-flop is much slower than permeation of dithionite (time scale of hours versus minutes). These studies were performed on liposomes with a much simpler lipid composition (generally, a single lipid class), with smaller liposomes, and with a more limited set of polyunsaturated species.

This long introduction to say that if we do not master all factors at play in the experiments shown in Figure 2C, the second point of this list (multi-lamellarity) is probably responsible for the lack of complete quenching. In the revised text, we now write:

”Note that the liposomes used in the dithionite experiments were the same as that used in the dynamin experiments (see Figure 1B) to allow a direct comparison between the two assays. […] Despite these limitations, these experiments suggest that the presence of two polyunsaturated acyl chains in phospholipids strongly compromise membrane impermeability to ions.”